# An Overview of Analytical Methodologies for Determination of Vancomycin in Human Plasma

**DOI:** 10.3390/molecules27217319

**Published:** 2022-10-28

**Authors:** Xin Cheng, Jingxin Ma, Jianrong Su

**Affiliations:** Clinical Laboratory Center, Beijing Friendship Hospital, Capital Medical University, Beijing 100050, China

**Keywords:** vancomycin, analytical method, human plasma, bioassay, immunoassay, chromatography

## Abstract

Vancomycin is regarded as the last resort of defense for a wide range of infections due to drug resistance and toxicity. The detection of vancomycin in plasma has always aroused particular concern because the performance of the assay affects the clinical treatment outcome. This article reviews various methods for vancomycin detection in human plasma and analyzes the advantages and disadvantages of each technique. Immunoassay has been the first choice for vancomycin concentration monitoring due to its simplicity and practicality, occasionally interfered with by other substances. Chromatographic methods have mainly been used for scientific research due to operational complexity and the particular requirement of the instrument. However, the advantages of a small amount of sample needed, high sensitivity, and specificity makes chromatography irreplaceable. Other methods are less commonly used in clinical applications because of the operational feasibility, clinical application, contamination, etc. Simplicity, good performance, economy, and environmental friendliness have been points of laboratory methodological concern. Unfortunately, no one method has met all of the elements so far.

## 1. Introduction

Vancomycin is a tricyclic glycopeptide antibiotic produced by the fermentation of Streptomyces orientalis that was isolated in soil samples from the jungles of Borneo, Indonesia in 1956 [1]. It inhibits bacterial synthesis by three main mechanisms: inhibition of the synthesis of peptidoglycan, alteration of the cell membrane permeability, and interference with RNA synthesis in the cytoplasm [2]. The antibacterial spectrum of vancomycin mainly includes aerobic Gram-positive bacteria such as Staphylococcus, Streptococcus, Enterococcus, Corynebacterium, Listeria, and Clostridium difficile [1]. Currently, vancomycin is primarily used for the treatment of infections due to methicillin-resistant Staphylococcus aureus (MRSA) and methicillin-resistant Staphylococcus epidermidis (MRSE), pseudomembranous enteritis due to Clostridium difficile, an alternative to β-lactam allergy, the prevention of endocarditis, and infections during prosthetic implantation [3].

Vancomycin is poorly absorbed in the digestive tract, and intramuscular injection can cause severe local pain and tissue necrosis. It is routinely administered intravenously via 5% dextrose or 0.9% saline. After injection, vancomycin is rapidly distributed to many body tissues, reaching effective therapeutic concentrations in the lung, heart, synovial fluid, peritoneal fluid, bone, and kidney. However, it cannot cross the blood–brain barrier in a non-inflammatory state. Its protein binding rate is about 55%. Moreover, in patients with normal renal function, more than 90% of the drug is excreted in the unchanged form via the kidneys [1,4]. Its pharmacokinetics is influenced by several factors including the patient’s age, body weight, serum albumin, urine pH, and combined medications [5]. The half-life of vancomycin is closely related to renal function, ranging from 6 h in those with normal renal function to 7 days in anuric patients [1]. 

Vancomycin is a time and concentration-dependent (AUC-dependent) antibiotic with a post-antibiotic effect. According to pharmacokinetic (PK)/pharmacodynamic (PD) theory, the evaluation indicator for therapeutic drug monitoring (TDM) of vancomycin is the area under the concentration–time curve (AUC)/minimum inhibitory concentration (MIC) ratio, while with a AUC/MIC ratio ≥ 400 (based on MIC ≤ 1 µg mL^−1^), vancomycin can achieve a clinical effect [4,6]. However, when MIC > 1 µg mL^−1^, it is recommended to switch to another drug [6]. Vancomycin has a narrow therapeutic window (effective concentration is close to toxic concentration). Insufficient drug concentration can easily lead to the development of bacterial resistance, and too high a concentration is prone to serious adverse effects on the body [7] such as nephrotoxicity, ototoxicity, hypotension, phlebitis, hypersensitivity reactions, red man syndrome, neutropenia, chills, fever, and interstitial nephritis. Therefore, it is necessary to perform a TDM for vancomycin. 

Since the first use of vancomycin, researchers have devoted themselves to detecting vancomycin in plasma by different methods such as the bioassay, radioimmunoassay (RIA), fluorescence polarization immunoassay (FPIA), the enzyme-multiplied immunoassay technique (EMIT), high-performance liquid chromatography (HPLC), liquid chromatography-tandem mass spectrometry (LC-MS), and other methods. 

To the best of our knowledge, this is the first review of analytical methodologies for the determination of vancomycin in human plasma, except for only one recent review that focused on vancomycin analytical methods in the last five years [8]. However, the review did not provide a visual and detailed introduction for each technique and lacked other crucial methods, especially immunoassay, which remains the most robust method in the clinic. Hence, we consulted and systematically collected almost all relevant research materials as early as 1968. However, recent studies have paid more attention to the role of the drug in particular populations such as renal failure patients, children, or the elderly, whose pharmacokinetics are abnormally influenced by biological conditions. There were still a few advanced research investigations exploited by intersecting disciplines such as nanomaterial and new probes. With the increased antibiotic-resistance pressure caused by drug abuse, there is an urgent need to detect antibiotics in a more sensitive and specific manner. 

## 2. Bioassay

Bioassays provide a more visual assessment of vancomycin concentrations based on the antibacterial activity of vancomycin in vitro. Bioassay methods for vancomycin were studied mainly in the 1980s (Table 1) [9,10,11,12,13,14]. The basic steps are as follows: uniform inoculation of an indicator bacterium on a suitable medium, punching holes in the medium with a punch and pouring in vancomycin solution [8], or soaking paper sheets in vancomycin-containing solution and sticking them on the medium [10,11,12,13,14]. The inhibition circle diameter is linearly related to the vancomycin concentration or its logarithm. The corresponding vancomycin concentration can be calculated from the standard curve and the inhibition circle diameter. The key in this method is the selection and preparation of the indicator bacterium and medium. Bacillus spp. is sensitive to most antibiotics and is often used as an indicator bacterium. Furthermore, the medium should neither affect the indicator bacterium’s growth nor the antibiotic activity [15]. Other factors such as the pore size, incubation time, and incubation temperature can also affect the diameter of the antibacterial coil. This method’s standard concentration range (0.8–80 µg mL^−1^) covers the routine dose concentration range of vancomycin, which meets the clinical needs. Moreover, it is cheaper compared to immunoassays and liquid chromatography. The bioassay results were consistent with those of FPIA, HPLC, RIA, and fluorescence immunoassay [9,13]. Nevertheless, the operational steps are more cumbersome than other methods such as immunoassay and overnight incubation, which further prolong the experimental operation time. Although investigators often repeat a single test ten times [9] or four times [10] in trials to avoid random error, the precision and accuracy are still not better than other methods. In addition, patients are often not mono-medicated in clinical settings. It is true that aminoglycosides can be inhibited by increasing the concentration of NaCl to 6.0% in the plate, and rifampicin does not affect the inhibitory activity of vancomycin. However, other drugs such as β-lactams, macrolides, and sulfonamides still affect the application of this method [9,12].

In conclusion, the method is limited in its clinical application due to the use of multiple drugs in clinical practice and the long operation time. Nevertheless, in primary hospitals without expensive instruments (e.g., immunoassay analyzer, liquid chromatograph), a bioassay is still recommended for assaying the drug concentration in plasma. However, further research is needed to inactivate the activity of other antibacterial drugs to reduce the interference of vancomycin detection. 

## 3. Immunoassay

Vancomycin has a molecular weight of 1449 Da, which does not stimulate the human body to produce relevant antibodies. Therefore, an immunoassay is not affected by anti-vancomycin antibodies produced by the human body [16]. Due to its small molecular weight, the competition method is the primary type of immunoassay for vancomycin. This method uses labeled vancomycin and free vancomycin in blood to compete for binding to the corresponding antibody; the more free-vancomycin is present in the blood, the less labeled vancomycin can bind to the antibody. The markers can be isotopes, fluorescence, or enzymes. According to the type of marker, these methods can be classified as the RIA, fluorescent immunoassay (FIA), and enzyme immunoassay (EIA).

### 3.1. RIA

Crossley et al. [14] and Fong et al. [16] detailed the process of the vancomycin radioimmunoassay. First, vancomycin is conjugated with bovine serum albumin and then injected intravenously into test rabbits to obtain antibodies; ^3^H acrylated or ^125^I iodinated vancomycin, respectively; the vancomycin standard solution or vancomycin in serum competed with the labeled vancomycin to bind antibodies, and the radioactive signal was recorded by scintillation spectrometry (^3^H) or γ-counter (^125^I). The method’s sensitivity was 0.04 ng mL^−1^ to the maximum, while that of the biological method was 0.8 µg mL^−1^. RIA was significantly more sensitive than the biological method. Moreover, vancomycin could be detected in the serum and urine after oral vancomycin administration, which has an advantage in terms of small patient sample size or studying vancomycin metabolism in the organism [16]. Moreover, the results corresponded well with those obtained by the biological methods, FPIA and HPLC, while it was slightly less accurate and precise [8,14]. The RIA results were high compared to the biological method. Some studies attributed this to the degradation of vancomycin at low pH [14]. Meanwhile, the isotope quenching should not be neglected in the experiment. Disadvantages of RIA are apparent [17]: preservation and waste disposal of radioisotope reagents; hazards to humans; relatively short shelf-life; the need for dilution before sample detection; the need to separate antibody-bound and free fractions before counting; and the fact that counters are expensive and not equipped in routine bacterial laboratories. Based on the above facts, RIA has not been routinely used in laboratories.

### 3.2. FIA

FPIA is the most widely used method in FIA. The method requires a fluorescence polarizer and a fluorescently labeled antigen as a tracer. Meanwhile, it needs to avoid interference from endogenous fluorescence [18]. The relevant principles were elucidated in a previous literature review [18,19]. The tracer competes with the analyte to bind the antibody, and the tracer binds the antibody with a significantly higher fluorescence polarization value than the free tracer. Eventually, the analyte concentration is inversely related to the detected fluorescence polarization value. Schwenzer et al. [20] first reported the measurement of vancomycin by the FPIA instrumental method (Abbott TDx) in 1983. This method correlated well with other methods (Table 2) [8,11,13,20,21,22,23], with a minimum detection limit of 0.6–2 µg mL^−1^. The detection range could cover the blood concentration of vancomycin after the regular dose. It is simpler and more rapid than other methods and is suitable for carrying out in the clinic [20,24]. The FPIA assay for vancomycin has high precision, and the results were higher than for the other methods [25]. Backes [26] et al. used HPLC to confirm that the presence of the crystalline degradation product (CDP-1) was responsible for the high FPIA results. CDP-1 has two isomers, CDP-1-M (major) and CDP-1-m (minor), both of which have no antibacterial activity but can cross-immunoreactive with anti-vancomycin antibodies, thus leading to high results, especially in patients with kidney injury [23,26]. Other factors such as the poor stability of standards [27], drug accumulation due to altered vancomycin pharmacokinetics in patients with kidney injury [28], and the use of sheep-derived polyclonal antibodies in the FPIA method [25] may contribute to the high FPIA results. For clinical decision-making, the results of FPIA were elevated by approximately 14%. However, there was no need to adjust the vancomycin treatment dosage, given that the results were still within the vancomycin treatment window [22]. In conclusion, as a quick and easy method for the laboratory detection of vancomycin blood levels, the method’s accuracy can be improved by increasing the frequency of the assay, the use of monoclonal antibodies, and proper preservation of the standards and drugs. Other methods are recommended for patients with renal injuries such as EMIT and HPLC. 

### 3.3. EIA 

EMIT is one of the more commonly used clinical methods in EIA and can be used to test samples using conventional biochemical analyzers and commercial kits rapidly. It is performed by competing for the enzyme-labeled semi-antigen with the analyte to bind the antibody. The activity of the enzyme-labeled semi-antigen will be lost after binding to the antibody. Finally, the analyte concentration will be determined based on the change in absorbance after the enzyme-catalyzed reaction [29,30]. EMIT and FPIA, commonly used in clinical laboratories, are often used for comparison by investigators. They correlated well with each other. Both had fine precision and accuracy, which can meet the needs of clinical laboratories [23,25,27,28]. However, a multicenter retrospective study from the United Kingdom showed that EMIT would be more prone to random errors occurring than FPIA [31]. FPIA is susceptible to high results due to CDP-1, as mentioned previously. Both methods are more straightforward and faster than other methods such as the bioassay and HPLC, and they are not as harmful as RIA, so they are widely used in clinical laboratories. Both require regular internal quality control and external quality assessment to ensure the precision and accuracy.

It should be noted that EIA is occasionally affected by endogenous cross-reactive substances such as rheumatoid factor, heterophilic antibodies, paraproteins, C-reactive protein, or unexplained substances affecting enzyme activity, leading to falsely elevated vancomycin measurements and treatment failure due to the underdosing of patients [32,33]. Laboratory workers can reduce the interference of other protein-like substances by polyethylene glycol precipitation or heat inactivation methods. Alternatively, they can further use HPLC or LC-MS/MS to detect vancomycin concentrations. When clinicians encounter results that are inconsistent with clinical outcomes, they should consider the presence of assay influencing factors and communicate with laboratory workers promptly to minimize the impact of false results on the patients’ treatment decisions.

## 4. LC

LC is an essential method for isolating and detecting vancomycin in plasma or serum. Table 3 reviews the methods and chromatographic analysis conditions for determining vancomycin in human plasma or serum [21,22,34,35,36,37,38,39,40,41,42,43,44,45,46,47,48,49,50,51,52,53,54,55,56,57,58,59,60,61,62,63,64,65]. Undeniably, LC can analyze different body fluids such as plasma, serum, urine, cerebrospinal fluid, alveolar lavage fluid, atrial fluid, and other drug components or drug solutions prepared with different solvents. The routine laboratory testing of blood drug concentration specimens is plasma or serum, so healthy human plasma or serum without antibiotic use is often used as a solvent for vancomycin analysis. Thus, the study results are closer to the actual situation, reducing the errors caused by the matrix effect. Routine steps in LC include sample pretreatment, selection of the internal standard, separation by the stationary phase and mobile phase, and detection.

### 4.1. Pretreatment

Drugs in blood exist in both bound and unbound forms by binding proteins or not. The unbound form (free form) in blood has antibacterial activity and free drug can be separated from the bound drug by certain means such as equilibrium dialysis, ultrafiltration, ultracentrifugation, on-line or off-line methods, and non-separative methods, as previously reviewed [66]. Protein-binding studies have used these methods to investigate the relationship between free and total vancomycin in plasma, aiming at exploring whether total vancomycin can be used to assess free vancomycin [67,68]. Unfortunately, both of the results were negative. In addition, the experimental conditions (molecular weight cut-off, centrifugal force and time, pH, temperature) may affect the isolation of free vancomycin by ultrafiltration [38]. However, studies on free vancomycin remain scarce despite its importance. Total vancomycin is routinely tested in the laboratory [67]. In the LC method, pretreatment is conducted to eliminate interference from plasma proteins and other macromolecular substances. All samples were physically separated by centrifugation, and most studies added precipitates to denature the protein [21,22,34,35,36,39,40,41,42,43,44,45,46,47,49,51,52,53,54,55,56,57,58,59,60,61,62,63,64]. In addition, a few studies isolated the proteins in a combined solid phase extraction (SPE) [21,59] manner. Precipitants include ACN, MeOH, TCA, and HClO_4_; the most commonly used are ACN and MeOH solutions. ACN and MeOH are often used as subsequent mobile phase components. Hence, neither increases the interference analyzed in chromatography, further reducing the matrix effect. There was only one recovery result for TCA and HClO_4_ as precipitants. Bijleveld et al. found a maximum recovery of 70% at 15–35% (*w*/*v*) TCA concentration. Combined with other extraction methods, the analyte could be purified efficiently. For example, SPE effectively separated analytes and interferents, even though vancomycin 0.05 µg mL^−1^ recovery could reach 87.6% [59]. The complex sample pretreatment process improves the purity of the analytes. However, it also increases the labor and economic costs, which is not conducive to detecting drug concentration. Aqueous ACN or MeOH solution for protein precipitation, then centrifugation, followed by supernatant extraction, is the most economical and convenient sample pretreatment mode. After sample pretreatment, the supernatant is evaporated by liquid nitrogen to a dry powder state and can be used for the next chromatographic step.

### 4.2. Internal Standard (IS)

Appropriate IS allows for monitoring of the sample pretreatment process, column injection volume, and even the evaluation of the calculated sample volumes. The amount of the target analyte can be assessed or calculated from the peak area of the IS and the peak area of the analyte. Although a number of studies [36,38,40,43,45,47,49,50] have not mentioned or used IS, these studies still obtained reliable findings. Most studies considered the inclusion of IS in the sample pretreatment. For the selection of IS, ultraviolet (UV) detection and mass spectrometry (MS) detection have different requirements. In UV detection, the peak overlap between the analyte, IS, and endogenous plasma should be avoided. The λmax of IS should be as close as possible to the λmax of the analyte and has good absorbance [34]. Hence, tinidazole [21], acetaminophen [34], 3-nitroaniline [35], zidovudine [37], ristomycin [39], caffeine [41], ketoprofen [42], and cefuroxime [44] can be used as IS for the vancomycin assays. The effect of the matrix ionization of analytes in MS detection could be best compensated using a stable isotope-labeled IS. Therefore, isotopically labeled vancomycin with similar extraction recovery, chromatographic characteristics, and ionization response to the desired analyte are the preferred choice of IS for MS detection for commercialization and the most applied vancomycin derivatives in the studies were vancomycin derivatives such as desmethyl vancomycin [51,52,54], desleucine vancomycin [58], and vancomycin-glycin [61], all of which have similar features to the isotope-labeled vancomycin. Others such as tobramycin [22], PABA [46], erythromycin [48], roxithromycin [53], linezolid [55], 10-hydroxycarbazepine [56], atenolol [59], 13C3-caffeine [60], polymyxin B [62], and kanamycin B [64] were confirmed to be applied as IS in the study.

### 4.3. Stationary Phase and Mobile Phase

Based on vancomycin polarity and molecular weight size, vancomycin is most often separated using the reversed-phase liquid chromatography mode [21,22,34,35,36,37,38,39,40,41,42,43,44,45,46,47,48,49,50,51,52,53,54,55,56,59,60,61,62,63,65], and a few studies have used hydrophilic interaction chromatography [57,58] and reverse ion exchange chromatography [64]. The difference between the various separation modes lies in the choice of stationary and mobile phases. The most commonly applied stationary phase in the reversed-phase liquid chromatography mode was a C_18_ column [22,34,35,37,38,40,41,42,43,44,45,46,47,48,49,50,51,52,53,54,55,56,60,61,62,63,65], and a few used the C_8_ column [21,36,50,59] and aminopropyl silica column [39]. In reversed-phase liquid chromatography mode, the most commonly used mobile phases for UV detection were acetonitrile and pH 2.5 buffer such as phosphate buffer [34,35,37,38,39,40,41,44,49,50], while MS detection was performed with acetonitrile and 0.1% formic acid [51,52,53,54,55,56,59,60,62,63,65]. HILIC hydrophilic interaction columns are hydrophilic and elute in the opposite order to reversed-phase liquid chromatography. This chromatographic mode retains highly polar and hydrophilic drugs well and is highly compatible with electrospray mass spectrometry detection [57]. This mode results in longer vancomycin retention times due to vancomycin polarity and hydrophilic effects. However, in the studies by Parker et al. [57] and Oyaert et al. [58], the vancomycin retention time was 2 min and 2.7 min, which effectively solved the problem of the long retention time of HILIC. In addition, Bijleveld et al. [64] used reverse ion exchange chromatography for vancomycin separation. The most important feature of this study was the addition of ionic pair 200 mM perfluorovaleric acid/130 mM ammonium acetate to the mobile phase, and other conditions were the same as those of reversed-phase liquid chromatography.

### 4.4. Detection

Various detection methods have been used including UV detection, MS detection, photodiode array (PDA), electrochemical detection (ECD), and fluorescence detection (FLD), with the first two being the most commonly used detection methods. Ghasemiyeh et al. [34] plotted a standard curve of vancomycin concentration versus the ratio of vancomycin peak area to acetaminophen peak area to obtain the vancomycin concentration, while Hu et al. [36] recorded the UV–Vis spectra for each retention time and obtained a matrix (elution time × wavelength) for each sample analyzed, which was mathematically separated using two trilinear decomposition algorithms. Neither of these studies used the detection method described above. The wavelength range in UV detection was 205–282 nm, with 240 nm being the most used detection wavelength [37,38,40,41,42,43,44,45,49,50]. MS detection detectors included triple quadrupole [22,52,53,54,55,57,58,60,61,62,63,64,65], Q-Trap [51,56], and Orbitrap [59], and the ion sources were all in ESI (+) mode. The MS assay (10–200 µL) generally used less sample than the UV assay (50–2000 µL), PDA (200 µL, 1000 µL) and fluorescence assay (500 µL). The amount of plasma was significant for certain special populations. For example, a method that requires 1000 µL of sampling is more difficult to implement for the TDM of children. Moreover, the sensitivity and linear range of MS and UV detection was comparable at 0.1–1 µg mL^−1^ (sensitivity range) and 0.1–100 µg mL^−1^ (linear range), respectively, except for one study that used the Q-Trap assay with a sensitivity of 1 ng mL^−1^. In addition, the sensitivities of the FLD and PDA assays could reach 2 ng mL^−1^ and 1 ng mL^−1^, respectively.

## 5. Other Methods

In addition to the above bioassay, immunoassay, and chromatography, the researchers are still constantly exploring new methods.

### 5.1. Spectrophotography

Earlier in 1968, Fooks et al. [69] determined the vancomycin concentration by colorimetric reaction based on the reaction of the phenol group in the structure of vancomycin with the Folin–Ciocalteau reagent, which can produce a blue reaction. Later, Fathalla et al. [70] used a similar principle to treat vancomycin hydrochlorides with nitrite to form a nitroso derivative, which was then measured spectrophotometrically or by polarization. Despite this method not being affected by other glycopeptides and vancomycin degradation products, other phenol group-containing substances may affect the interpretation of the results. This method of vancomycin detection has not been further investigated by subsequent investigators

### 5.2. Micellar Electrokinetic Capillary Chromatography (MEKC)

Toshihiro et al. [71] used the MEKC method to detect the serum vancomycin concentration. The method uses direct injection of the sample without extraction pretreatment before capillary electrophoresis, followed by light-emitting diode detection to obtain the electrophoretic signal and for protein separation. The electrophoresis buffer was 100 mM SDS. The linear range of vancomycin detection by the MEKC method was 0–100 µg mL^−1^ and the detection limit was 0.1 µg mL^−1^. Another study used the MEKC method to determine cefepime and vancomycin in plasma and the cerebrospinal fluid of patients with bacterial meningitis [72]. To achieve proper resolution, the electrophoresis buffer was 200–300 mM SDS and 18% methanol. Given that current interruption may occur with high SDS concentrations, Wang et al. [73] further modified the buffer to 25 mM borate buffer, 50 mM SDS, and 2% sulfobutyl-β-cyclodextrin (pH 9.5) and successfully performed the daily monitoring of serum vancomycin concentrations for patients with peritoneal dialysis-associated peritonitis. The method does not require complex sample processing, replaces expensive columns with economical capillaries, and eliminates the need for a mobile phase, reducing organic solvent consumption and contaminants, and the high sensitivity and wide linear range allow the method to be used for the TDM of vancomycin.

### 5.3. Nanophase Materials

The combination of novel materials and chemiluminescence can also be used to detect the vancomycin concentration. CuO nanomaterials can act as signal amplifiers for chemiluminescence while vancomycin can inhibit the chemiluminescence signal generated by lumino-H_2_O_2_-CuO nanosheets. Khataee et al. [74] combined the principle to study a CuO nanosheet amplified flow injection chemiluminescence system capable of detecting vancomycin, which is inversely proportional to the chemiluminescence intensity. The method detected vancomycin in the linear range of 0.5–18.0 µg mL^−1^ and 18.0–40.0 µg mL^−1^ with a detection limit of 0.1 µg mL^−1^, and the linear range and the minimum detection limit could cover the routine vancomycin monitoring concentration. Atal et al. [75] obtained the vancomycin concentration based on the change in potential by a redox reaction of a copper-containing metal-organic material bound to vancomycin. The linear range (1–500 nM) and the minimum detection limit (1 nM) of this method are more suitable for vancomycin detection in urine compared to serum.

### 5.4. New Probe

Novel probes have been investigated to increase the diversity of vancomycin assays. Bai et al. [76] investigated a highly specific and stable peptide-functionalized cantilever array probe to detect the vancomycin concentration. The sensor structure included cysteine (Cys-), a spatial linker (-Gly-Gly-Gly-), and a molecular recognition ligand (-L-Lys-D-Ala-D-Ala), and binding to vancomycin caused a change in stress on the surface of the cantilever beam and deflected the sensor cantilever beam. The method had a detection limit of 0.2 µg mL^−1^ and the structure bonded specifically to vancomycin, ensuring high specificity of the method. Another study [77] designed a specific probe by binding two peptide chains of a dimeric derivative of the vancomycin-binding peptide L-Lys-D-Ala-D-Ala to each of the fluorescent dansyl chloride groups, which were combined with microdialysis sampling techniques to achieve continuous monitoring of vancomycin in rabbits. Similarly, Philippe et al. [78] designed an electrochemical aptamer-based sensing platform that enabled the continuous monitoring of vancomycin in a mouse model. Tao et al. [79] integrated coumarin and fluorescein as a dual fluorescent reporter group and vancomycin-binding peptide D-Ala-D-ala to synthesize a novel selective antibiotic vancomycin probe. These studies provide an alternative idea for the detection of vancomycin and other substances, and further confirmation of its clinical feasibility is needed.

### 5.5. Molecularly Imprinted Polymer Nanoparticles (MIPs)

The MIP technique, similar to ELISA, has also been used for vancomycin detection in recent years [80,81]. The technique consists of two main steps: the screening of MIPs and immobilization in the enzyme plate, and competition between vancomycin and enzyme-labeled vancomycin for binding MIPs. The linear range of the method (0.001–70 nM) and the minimum detection limit (2.5 pM) were far superior to other methods [80].

In addition, both fluorescence [82] and infrared spectroscopy [83] based on vancomycin’s properties were confirmed to be useful for vancomycin detection.

## 6. Summary and Outlook

Antimicrobial drug concentration monitoring has been an important factor affecting the rational use of drugs. Accurate antimicrobial drug results can avoid ineffective treatment and resistance problems caused by insufficient drug concentrations and damage to the body caused by high drug concentrations. Therefore, researchers have been searching for new methods to meet the clinical needs. This review was aimed at seeking a suitable method or thinking for drug concentration detection by summarizing various methods of vancomycin detection in human plasma. In general, an ideal method should possess the advantages of easy operation, accuracy, economy, and no pollution to the environment.

### 6.1. Simplicity

Given the complex and laboring task, simplicity is a vital factor among all considerations for laboratory workers. Therefore, the new method should be as simple as possible under the premise of accuracy. As for bioassay, LC, and other novel methods, time-consuming operations and complex procedures limit their routine practice. Especially in LC, which has been validated by many studies, elaborate pretreatment and professional post-experimental analysis put a heavy burden on laboratory technicians. Simplifying relative procedures, for example, adding an autosampler and strong software, will enable the method to become more promising. In contrast, FPIA and EMIT, both of which have commercialized reagents and are easy to operate, are the most commonly used methods for vancomycin detection in the clinic.

### 6.2. Good Performance

From spectrophotometry to novel methods, accuracy and sensitivity, two core parameters to be compared by various methods, were the first factors in the experiment. Up to now, HPLC and LC-MS are the gold standards of vancomycin detection methods due to higher sensitivity, better specificity, a wide linear range of detection, and a small amount of sample in comparison with the bioassay and immunoassay, which have a very broad application prospect. In contrast, the immunoassay is destined to be inevitably affected by other factors such as other immunoglobulins and CDP-1. Hence, HPLC and LC-MS are used as reference methods for false results of the immunoassay. Although the application of new materials and probes has enriched the diversity of antibiotic detection and provided new detection ideas, these novel methods tend to be more sensitive but lack clinical verification. These methods are less studied and need to be further applied to confirm their clinical value.

### 6.3. Economy

Nowadays, laboratories are equipped with expensive instruments and software for routine. However, the economy has been measured in vancomycin detection, especially in need of frequent TDM. In the 1980s, bioassays were the mainstream methods despite the occurrence of HPLC and immunoassays. It provides a more visual assessment of vancomycin concentrations based on the antibacterial activity of vancomycin in vitro. More importantly, laboratory materials such as culture medium and the disc diffusion test are economical and practical for primary hospitals that lack expensive apparatus. Meanwhile, expensive detection reagents make the immunoassay criticized, forcing researchers to look for new ways as a succedaneum.

### 6.4. Environmental Friendliness

We cannot ignore the influence caused by reagents and waste. In the review, despite the excellent sensitivity of RIA, the problem of radioactivity leads to the limitation of its clinical application. Similarly, LC-MS can achieve timely and high-throughput detection of a variety of antibacterial drugs, but toxic organic solvents should be considered for replacement by green counterparts.

## Figures and Tables

**Table 1 molecules-27-07319-t001:** Bioassay for vancomycin.

Medium	pH	Indicator Organism	Standard Range (µg mL^−1^)	Well Size (mm)	Incubation Time (h)	Incubation Temperature (°C)	Ref.
Tryptic soy agar medium	none	Bacillus globigii	5–80	2.5	18	35	[9]
MSA	7.3	Bacillus subtilis (W23)	0.8–50	6	10–12	35–37	[10]
MSA	7.3	Bacillus subtilis ATCC 6633	0.8–50	6	16–18	36	[11]
Antibiotic medium no. 5	8.0	Bacillus subtilis ATCC 6633	10–40	None	4–18	37	[12]
MSA	7.3	Bacillus subtilis ATCC 6633	0.8–50	None	16–18	35	[13]
Heart infusion agar medium	5.5	Bacillus subtilis ATCC 6633	4–32	6.4	8	37	[14]

Abbreviations: MSA: minimal salts agar. Ref.: reference.

**Table 2 molecules-27-07319-t002:** Correlation analysis between FPIA and other methods.

Detection Range (µg mL^−1^)	Other Methods	Correlation Coefficient	Ref.
5–85	Bioassay	0.973	[8]
	HPLC	0.977	
	RIA	0.967	
	FIA	0.918	
5–80	Bioassay	0.985	[12]
0.8–50	Bioassay	0.777	[13]
	HPLC	0.999	
0.6–100	LC	0.980	[20]
	RIA	0.975	
0.5–75	HPLC	0.964	[21]
0.1–100	LC-MS/MS	0.943	[22]
5–50	HPLC	0.939	[23]
	EMIT	0.979	

Abbreviations: LC: liquid chromatography. Ref.: reference.

**Table 3 molecules-27-07319-t003:** Summary of the plasma measurement method and analysis of vancomycin.

Method	Plasma/Serum	IS	Precipitant	Column	Mobile Phase	Detector (nm)	tR (min)	Sensitivity LOD/LOQ	Lin. Range	Ref.
HPLC	500 µL	Tinidazole	MeOH	Hypersil BDS C8	5 mM potassium dihydrogen phosphate buffer (pH 2.8)-ACN	UV; 282	7.6	LOQ: 0.5 µg mL^−1^	0.5–75 µg mL^−1^	[21]
LC-MS/MS	50 µL	Tobramycin	33% TCA	RP BEH C18	A: 2 mM ammonium acetate, 0.1% FA in 5% ACN; B: 2 mM ammonium acetate, 0.1% FA in 95%MeOH	MS/MS, ESI (+)	None	LOQ: 0.1 µg mL^−1^	0.1–100 µg mL^−1^	[22]
HPLC	500 µL	Acetaminophen	70% HClO_4_	Azura C18	Phosphate buffer (30 mM, pH of 2.2) and ACN (86:14% *v*/*v*)	None	5.5	LOD: 0.3 µg mL^−1^; LOQ: 1 µg mL^−1^	1–30 µmL^−1^	[34]
HPLC	1000 µL	3-Nitroaniline	MeOH	μBondapak C18	A: triethylamine buffer, ACN, tetrahydrofuran (92:7:1); B: was the same solvent with 70:29:1 ratio	PDA; 205	None	1 ng mL^−1^	1–10 ng mL^−1^	[35]
HPLC-DAD	2000 µL	None	none	XDB-C8	Methanol and 0.1 M disodium hydrogen phosphate buffer (40/60 *v*/*v* %)	UV–Vis	None	LOD: 0.32 µg mL^−1^	None	[36]
HPLC	50 µL	Zidovudine	none	Supelcosil C18	20 mM ammonium acetate/FA buffer (pH 4.0): methanol 88:12 (*v*/*v*)	UV; 240	4.0	LOQ: 1 µg mL^−1^	1–100 µg mL^−1^	[37]
HPLC	300 µL	None	none	CXBridge C18	Phosphate buffer; ACN	UV; 240	None	None	None	[38]
HPLC	50 µL	Ristomycin	15% HClO_4_	Microsorb-MV-NH_2_	62% acetonitrile, 38% sodium phosphate buffer (pH 7.0)	UV; 225	None	LOD: 0.32 µg mL^−1^	1–100 µg mL^−1^	[39]
HPLC	200 µL	None	70% HClO_4_	Nucleodur C18	NH_4_H_2_PO_4_ (50 mM, pH 2.2)-CAN (88:12, *v*/*v*)	UV; 205	8.1	LOD: 0.25 µg mL^−1^	0.25–60 µg mL^−1^	[40]
HPLC	200 µL	Caffeine	ACN	Spherisorb C18 ODS	0.05 M ammonium phosphate buffer with 11% ACN	UV; 240	13.7	LOD: 0.1 µg mL^−1^	1–80 µg mL^−1^	[41]
UHPLC	1000 µL	Ketoprofen	ACN	Hypersil GOLD C18	0.1% FA, ACN	UV; 215	2.96	LOD: 0.01 µg mL^−1^	0.36–20 µg mL^−1^	[42]
HPLC	100 µL	None	MeOH, ACN	Accucore C-18	A: 0.1% TFA; B: ACN: Milli-Q water 40:60 (*v*/*v*) with 0.1% TFA	UV; 240	None	LOQ: 2 µg mL^−1^	None	[43]
HPLC	200 µL	Cefuroxime	ACN	Supelcosil^TM^ LC-18	0.075 M acetate buffer, pH 5.0, and ACN (92:8 *v*/*v*)	UV; 230	17.4	LOD: 0.17 ± 0.01 µg mL^−1^	0.4–100 µg mL^−1^	[44]
HPLC	100 µL	None	MeOH, TCA	CLC-ODS	0.05 M phosphate Buffer, MeOH, ACN	UV; 240	None	LOQ: 0.4 µg mL^−1^	0.4–80 µg mL^−1^	[45]
UPLC	200 µL	PABA	ACN	Acquity UPLC BEH C18	0.005 M KH_2_PO_4_ buffer (pH 2.5), ACN	PDA; 230	2.6	LOQ: 1 µg mL^−1^	1–100 µg mL^−1^	[46]
HPLC	100 µL	None	MeOH	Kromasil C18	ACN-sodium phosphate buffer (pH 7) (12:88)	ECD	None	LOD: 0.5 µg mL^−1^; LOQ: 1 µg mL^−1^	5–100 µg mL^−1^	[47]
HPLC	500 µL	Erythromycin	none	μBondapak C18	A: 5 mM KH_2_PO_4_, MeOH	FLD; Ex: 225, Em:258	16.3	LOD: 2 ng mL^−1^; LOQ: 5 ng mL^−1^	5–1000 ng mL^−1^	[48]
HPLC	1000 µL	None	HClO_4_	Nucleosil RP-18	0.005 M KH_2_PO_4_ (pH 2.8), ACN 90:10 (*v*/*v*)	UV; 229	5	LOD: 0.2 µg mL^−1^; LOQ: 1 µg mL^−1^	1–100 µg mL^−1^	[49]
HPLC	none	None	None	SPS octyl-C8	0.1 MNaH_2_PO_4_ buffer-CAN (95/5, *v*/*v*%)	UV; 240	None	LOD: 0.5 µg mL^−1^	0–100 µg mL^−1^	[50]
HPLC-Q-Trap-MS	100 µL	Norvancomycin	ACN	ZORBAX SB-C18	Water (containing 0.1% FA, *v*/*v*) and ACN (containing 0.1% FA, *v*/*v*)	Q-Trap, ESI (+)	None	LOD: 0.3 ng mL^−1^; LOQ: 1.0 ng mL^−1^	1–2000 ng ml^−1^	[51]
UPLC-MS/MS	200 µL	Norvancomycin	ACN	Acquity UPLC BEH C18	ACN, 0.1% FA	MS/MS, ESI (+)	None	LOQ: 1.0 µg mL^−1^	1–100 µg mL^−1^	[52]
UPLC-MS/MS	100 µL	Roxithromycin	ACN	Acquity UPLC BEH C18	ACN, 0.1% FA	MS/MS, ESI (+)	None	LOD: 0.02 µg mL^−1^; LOQ: 0.05 µg mL^−1^	0.05–10 µg mL^−1^	[53]
UPLC-MS/MS	50 µL	Norvancomycin	ACN	Acquity BEH C18	ACN, 0.1% FA	MS/MS, ESI (+)	None	LOQ: 0.25 µg mL^−1^	*Y* = 0.133 *x* − 0.00823 (*r =* 0.9980)	[54]
UPLC-MS/MS	50 µL	Linezolid	ZnSO4, ACN	Hypersil GOLD aQ C18	ACN, 0.1% FA	MS/MS, ESI (+)	1.28	LOQ: 0.1 µg mL^−1^	0.1–128 µg mL^−1^	[55]
LC-MS/MS	50 µL	10-hydroxycarbazepine	ACN	Zorbax SB-C18	ACN, 0.1% FA (5:95, *v*/*v*)	Q-Trap, ESI (+)	None	LOQ: 1 µg mL^−1^	1–100 µg mL^−1^	[56]
LC-MS/MS	10 µL	^2^H12-vancomycin	ACN	SeQuant zic-HILIC	A: 30% ACN, 10% acetone and 60% of 0.1% FA in water, *v*/*v*/*v*; B: contained 70% ACN, 10% acetone and 20% of 0.1% FA in water, *v*/*v*/*v*	MS/MS, ESI (+)	2	LOQ: 1 µg mL^−1^	1–100 µg mL^−1^	[57]
LC-MS/MS	40 µL	Vancomycin-des-leucine	ACN	Acquity UPLC BEH HILIC	ACN, 0.1% FA	MS/MS, ESI (+)	2.7	LOQ: 0.3 µg mL^−1^	0.3–100 µg mL^−1^	[58]
LC-MS/MS	200 µL	Atenolol	MeOH	ACE-3-C8	ACN, 0.1% FA (1:9)	Orbitrap, ESI (+)	3.75	LOD: 0.001 µg mL^−1^; LOQ: 0.05 µg mL^−1^	0.05–10 µg mL^−1^	[59]
LC-MS/MS	25 µL	Caffeine-^13^C3	MeOH	Acquity UPLC BEH C18	ACN, 0.1% FA	MS/MS, ESI (+)	None	LOD: 0.5 µg mL^−1^; LOQ: 2 µg mL^−1^	0.002–50 µg mL^−1^	[60]
LC-MS/MS	75 µL	Vancomycin-glycin	TCA	Fortis C8	A: aqueous FA (0.1% *v*/*v*); B: MeOH containing 0.1% FA (0.1% *v*/*v*)	MS/MS, ESI (+)	9.8	LOQ: 1 µg mL^−1^	1–84 µg mL^−1^	[61]
UPLC–MS/MS	100 µL	Polymyxin B	ACN	Kinetex C18	ACN, 0.1% FA	MS/MS, ESI (+)	1.62	LOD: 1.1 ng mL^−1^; LOQ: 0.5 µg mL^−1^	0.5–100 µg mL^−1^	[62]
LC-MS/MS	50 µL	PIP-d6, MER-d3, CEF-d3	ACN	Eclipse Plus C18	25 mM FA, ACN	MS/MS, ESI (+)	1.94–2.02	none	None	[63]
LC-MS/MS	25 µL	Kanamycin B	TCA	Thermo Scientific Hypurity Aquastar	A: H_2_O, B: ACN100%, C: perfluoropen tanoic acid (200 mM)/ammonium acetate (130 mM) in H_2_O	MS/MS, ESI(+)	3.02	LOQ:1 µg mL^−1^	1–100 µg mL^−1^	[64]
LC-MS/MS	50 µL	Vancomycin-d12	10% TCA	Accucore^TM^ Polar Premium	A: 0.1% FA in water; B: 0.1% FA in ACN	MS/MS, ESI (+)	1.4	LOQ: 1 µg mL^−1^	1–100 µg mL^−1^	[65]

Abbreviations IS: internal standard; Lin.: linearity; tR: retention time; LOD: limit of detection; LOQ: limit of quantification; Ref.: reference; UV: ultraviolet; MS: mass spectrometry; ACN: acetonitrile; MeOH: methanol; TCA: trichloroacetic acid; TFA: trifluoroacetic; acid; PABA: P-aminobenzoic acid; FA: formic acid; ESI (+): electrospray ionization in positive ion mode; FLD: fluorescence detector; PDA: photodiode array; ECD: electrochemical detection; Ex: excitation length; Em: emission length; A: mobile phase A; B: mobile phase B; PIP: piperacillin; MER: meropenem; CEF: cefepime.

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
