# Peer review of "An Overview of Analytical Methodologies for Determination of Vancomycin in Human Plasma"

_molecules, 2022, doi:10.3390/molecules27217319_

Round 1

Reviewer 1 Report

In this article the authors overview the literature on analytical methods for the determination of vancomycin. The authors propose an evaluation of the advantages and disadvantages of each of the methods presented.

1.   Even though the authors put a summary and a conclusion at the end of the article to summarize, according to the criteria (simplicity, performance, economy, and environmental criteria), the methods that best meet these criteria. It is regrettable that the authors do not conclude by indicating their opinion towards an evolution or not. For example, not long ago, immunoassay methods were the majority for the determination of immunosuppressant drugs but they are gradually being replaced by LC/MS² methods in medical analysis laboratories. Do the authors foresee this type of evolution for the determination of vancomycin (or whether no) and in which form (multiplexed or single methods)?

2.       Page 3 line 111 : The authors state “So immunoassay is not affected by antibodies produced by the human body”. But a few lines later (Page 5 lines187) most interferences were found “It should be noted that EIA was occasionally affected by endogenous cross-reactive substances such as rheumatoid factor, heterophilic antibodies, paraproteins, C-reactive protein, or unexplained substances affecting enzyme activity, leading to falsely elevated vancomycin measurements and treatment failure due to underdosing of patients”. Why this discrepancy?

3.       The authors do explain what is required as therapeutic monitoring performance. Recall the desired therapeutic range. There is no need to obtain a sensitivity of 0.001 µg/ml for therapeutic zones around 30 µg/ml. On the other hand, a method with a high sensitivity in general allows to decrease the volume of sampling which is not discussed by the authors. For example, a method that requires 1000 µl of sampling (ref 42, 49…) is more difficult to implement for children TDM.

4.       At no time do the authors compare the repeatability and reproducibility of the methods presented. Yet these elements are part of the performance criteria of an analytical method.

5.       Page 5 line 184 : “Both require regular indoor quality control and inter-room quality assessment to ensure precision and accuracy.” Other methods are not concerned with environmental quality criteria? Can the authors explain why?

6.       Page 6 line 244 : “However, stable isotope-labeled vancomycin is not yet available,…” False statement because in table 3 ref 57 and 65 the methods use 2H12 Vancomycine which is an isotope. Homogenize chemical names 2H12 Vancomycin et Vancomycin d12 it’s the same product.

7.       Putting in the same paragraph the triple quadripole, Q-trap and orbitrap methods poses a problem because they do not have the same performance in terms of repeatability and sensitivity. Develop these aspects. Economy is also different.

8.       Table 2 is poorly constructed we can’t see the lines.

9.       International units are L, not ml. µg/ml becomes mg/L and sometimes units are in µg/ml other times in µg ml-1 see ug ml-1 homogenize units

Author Response

Please see the attachment。

Round 2
